# Konjac Ceramide (kCer)-Mediated Signal Transduction of the Sema3A Pathway Promotes HaCaT Keratinocyte Differentiation

**DOI:** 10.3390/biology11010121

**Published:** 2022-01-12

**Authors:** Seigo Usuki, Noriko Tamura, Tomohiro Tamura, Kohei Yuyama, Daisuke Mikami, Katsuyuki Mukai, Yasuyuki Igarashi

**Affiliations:** 1Lipid Biofunction Section, Frontier Research Center for Advanced Material and Life Science, Faculty of Advanced Life Science, Hokkaido University, Sapporo 001-0021, Japan; kyuyama@pharm.hokudai.ac.jp (K.Y.); dmikami.f@frontier.hokudai.ac.jp (D.M.); mukai@gunma-u.ac.jp (K.M.); yigarash@pharm.hokudai.ac.jp (Y.I.); 2National Institute of Advanced Industrial Science and Technology (AIST), Sapporo 062-8517, Japan; n-tamura@aist.go.jp (N.T.); t-tamura@aist.go.jp (T.T.); 3R & D Headquarters, Daicel Corporation, Tokyo 108-8230, Japan

**Keywords:** ceramide, konjac, keratinocyte, differentiation, neuropilin1

## Abstract

**Simple Summary:**

Konjac ceramide (kCer) is a unique molecular species of plant-type ceramide, and is a potential Sema3A-like ligand of Nrp1. kCer suppresses histamine-stimulated cell migration of HaCaT keratinocytes. This effect of kCer is not due to histamine-activated GPCRs, but rather to Sema3A-Nrp1 receptor binding. The present study focused on the ability of kCer to induce cell differentiation, in addition to its anti-migratory effects. We demonstrated that the effects of kCer on cell migration and cell differentiation are perpetuated by a cascade of crosstalk between pathways downstream of Nrp1 and GPCR in HaCaT cells.

**Abstract:**

Histamines suppress epidermal keratinocyte differentiation. Previously, we reported that konjac ceramide (kCer) suppresses histamine-stimulated cell migration of HaCaT keratinocytes. kCer specifically binds to Nrp1 and does not interact with histamine receptors. The signaling mechanism of kCer in HaCaT cells is also controlled by an intracellular signaling cascade activated by the Sema3A-Nrp1 pathway. In the present study, we demonstrated that kCer treatment induced HaCaT keratinocyte differentiation after migration of immature cells. kCer-induced HaCaT cell differentiation was accompanied by some features of keratinocyte differentiation markers. kCer induced activating phosphorylation of p38MAPK and c-Fos, which increased the protein levels of involucrin that was the latter differentiation marker. In addition, we demonstrated that the effects of both kCer and histamines are regulated by an intracellular mechanism of Rac1 activation/RhoA inhibition downstream of the Sema3A/Nrp1 receptor and histamine/GPCR pathways. In summary, the effects of kCer on cell migration and cell differentiation are regulated by cascade crosstalk between downstream Nrp1 and histamine-GPCR pathways in HaCaT cells.

## 1. Introduction

Ceramides have been extensively used as supplements to prevent lifestyle-related diseases such as dry skin [1]. Ceramides are comprised of a sphingosine and a fatty acid residue and function primarily as intramolecular components of membrane lipid molecules such as ganglioside, sphingomyelin, and glucosylceramide (GlcCer). Free ceramides are also present on the mammalian cell membrane [2]. On the other hand, in plant tissue, free ceramide is composed of glycosylinositol phosphoceramide (GIPC) and GlcCer [3]. The remarkable differences in the molecular structures of animal and plant ceramides is attributed to a diversity of long-chain bases (Figure 1B). Molecular species of kCer are characterized by the combination of long-chain bases (d18:2, t18:1) and 2-hydroxyl fatty acids of C16:0, C18:0, C20:0, C22:0, C23:0, and C24:0 (Figure 1C) [4].

With the exception of citrus fruit, free ceramides represent a very small fraction of the lipid composition of plants [6]. Konjac (*Amorphophallus konjac*, K. Koch) is a food plant rich in GlcCer. Konjac GlcCer (kGlcCer) suppresses transepidermal water loss, and is used as a health food and in cosmetics [7]. We successfully prepared konjac-derived free ceramide (kCer) by deglucosylation of kGlcCer using endoglycoceramidase I (EGCase I) [4] (Figure 1A). 

Our prior study [4] demonstrated that kCer similarly inhibited neurite outgrowth, which was accompanied by changes in cell morphology to a spindle shape and short neurite retraction due to competition between Sema3A and nerve growth factor (NGF) [8]. We demonstrated that kCer binds the Sema3A receptor neuropilin1 (Nrp1) and can activate the Sema3A signaling cascade as a Sema3A-like ligand [9,10]. Nrp1 is also expressed as a Sema3A receptor in keratinocytes in vitro and in vivo [11,12,13]. We found that the Sema3A-like ligand function of kCer inhibits HaCaT keratinocyte migration [12]. Contrastingly, histamine enhances immature keratinocyte migration [14]. Histamine-induced cell migration is inhibited by both kCer and Sema3A. kCer is not associated with cell surface histamine receptor H1R (or H4R) binding in the histamine signaling pathway. 

HaCaT cells are able to reversibly convert between differentiated and basal states upon changes in extracellular calcium concentration. HaCaT cells are thus a useful in vitro keratinocyte differentiation model, in which cell culture medium calcium levels can be adjusted to regulate differentiation. Here, we used two culture conditions, short-term and long-term culture under low calcium concentration conditions [15]. In the present study, we demonstrated that HaCaT cell differentiation occurs after cell migration in cells treated with kCer. We hypothesize that kCer-induced cell differentiation could occur via signaling crosstalk mechanisms in pathways downstream of Sema3A/Nrp1 and histamine-activated G-protein coupled receptors (GPCRs).

## 2. Materials and Methods

### 2.1. Materials

*Antibodies.* The following antibodies were used for Western blot analysis: anti-p38MAPK (Bioss, bs-0637R, Woburn, MA, USA), anti-p-p38MAPK (phosphor Thr180/Tyr182, GeneTex, GTX133460, Irvine, CA, USA), anti-c-Fos (Abcam, ab222699, Cambridge, UK), anti-p-c-Fos (Abcam, ab27793), anti-involucrin (GeneTex, GTX1166012), and β-actin (FujiFilm, 010-27841, Tokyo, Japan).

*Chemicals.* The following chemicals were used for relevant inhibitor analysis: Rho pathway inhibitor I Rho kinase (ROCK) inhibitor Y-27632 (Cytoskeleton, Inc., CN069, Denver, CO, USA), Rac-1 inhibitor NSC237669 (AXN, Axon1578, Scottsdale, AZ, USA), p38 MAPK inhibitors SB203580 (AdipoGen, AG-CR1-0030, San Diego, CA, USA), BIRB796 (Selleck, S1574, Pittsburgh, PA, USA), and VX745 (Selleck, S1458). 

*Low Ca*^2+^*serum medium (L-Ca medium).* To prepare Ca^2+^ free-fetal bovine serum (FBS), 10 mL FBS was incubated with 2 g Chelex 100 chelating resin (Bio-Rad, Beerkeley, CA, USA) with shaking for 4 h at 4 °C. The mixture was centrifuged, and the supernatant preparation was agitated with fresh resin (2 g) overnight at 4 °C. The mixture was centrifuged, and the supernatant used for Ca^2+^-free FBS. Low-Ca^2+^ serum medium (0.02 mM) was prepared by adding Ca^2+^ aqueous solution, 10mL/L penicillin-streptomycin solution (WAKO Co., Osaka, Japan), and GlutaMAX^TM^I (×100, Gibco; Thermo Fisher Scientific, Inc., Waltham, MA, USA) to Ca^2+^-free FBS and Ca^2+^-free-Dulbecco’s modified Eagle’s Medium (DMEM, 21068-028). High-Ca^2+^ serum medium (2.8 mM, H-Ca medium) was prepared using regular DMEM containing high Ca^2+^ (1.8 mM) supplemented with Ca^2+^ aqueous solution, 10% FBS1, and penicillin-streptomycin/GlutaMAX™I. The Ca^2+^ concentration was measured using a calcium assay kit (ab102505).

### 2.2. kCer Preparation

kCer was prepared by modifying a method previously developed by our group [4]. Briefly, konjac GlcCer (kGlcCer, Nagara Science Co., Ltd., NS170303, Gifu, Japan) was subjected to two repetitions of a 16 h incubation with endoglycoceramidase (EGCase I) [4] followed by a Bligh-Dyer extraction [16]. kCer was analyzed by TLC using chloroform: methanol: acetic acid (65:10:0.1 *v*/*v*) as solvent. After development, TLC plates were sprayed with 10% cupuric sulfate in 8% phosphoric acid, and heated at 180 °C. The TLC image was obtained using a photoscanner and analyzed with JustTLC software ver. 4.0 (SWEDAY, Sodra Sanby, Sweden). Using the procedure described above, kCer was confirmed by purity and yield >95%.

### 2.3. Short-Term L-Ca Medium HaCaT Migration Model

HaCaT cells were first cultured in H-Ca medium and maintained in an incubator under 5% CO_2_ at 37 °C. The short-term cell culture was used for the HaCaT cell migration assay, with incubation for 48 h in L-Ca medium. After 48 h culture in L-Ca medium, involucrin expression was minimal, similar to cells cultured in undifferentiated conditions. A cell migration assay was performed using a transwell chamber plate (6.5 mm D.I., 8.0 μm pore size, polycarbonate membrane; Transwell Permeable Supports 3422, Corning Inc., New York, NY, USA) as previously described [12]. Cells subjected to short-term cell culture as described above were seeded at a density of 2 × 10^4^ cells/well in the upper chamber supplemented with 200 μL H-Ca medium, while 750 μL H-Ca medium with or without experimental reagents (Sema3A, kCer, or histamine) was added to the lower chamber compartment. After 48 h incubation at 37 °C, the unmigrated cells in the upper chamber were gently removed with a cotton swab, and the cells that had migrated to the lower compartment of the membrane were fixed with cold absolute methanol for 10 min and stained with GIEMSA’S AZUR EOSIN Methylene Blue solution (109,204, Merck, Darmstadt, Germany) as described previously [17].

### 2.4. Long-Term L-Ca Medium HaCaT Differentiation Model 

To obtain completely undifferentiated cells, cells were cultured in L-Ca medium for 3 weeks, with medium changed once weekly [15]. Cells were washed once in Ca^2+^ and Mg^2+^-free phosphate buffered saline (PBS) followed by trypsinization with 0.25% trypsin-EDTA solution. The trypsinization was stopped by addition of L-Ca medium. The cells were plated on 96-well microplate coated with 0.01% poly L-lysine and treated with H-Ca medium. Alternatively, cells were treated with L-Ca medium containing kCer or other ceramides. Immunofluorescence was used to assess the differentiation state of HaCaT cells by measuring the protein level of p-p38MAPK with a cell-based translocation assay kit (BVN, K965-100).

### 2.5. RNA Interference and Transfection

Gene silencing was performed using small interfering RNA (siRNA). Specific siRNAs (Silencer^®^Select, Ambion) were used: sense strand sequence 5′-GGUGUAUCAGAUGCUCAAAtt -3′ (p38δ), 5′-CCUAAAACCUAGUAAUCUAtt-3′(p38α), 5′-CAAACCAUAUCUUACCGUUtt-3′ (phospholipase C β1), and 5′-GGCUGUACUUCGUCAUGGAtt-3′ (protein kinase C α). To compare the efficiency of specific siRNA knockdowns, scrambled siRNA (medium GC Duplex; Invitrogen) was used as a negative control. siRNA transfection was conducted by transfecting 10 nM siRNA with Lipofectamine 2000 reagent mixed with Opti-MEM (Invitrogen), followed by incubating for 5 min at room temperature to allow complexation. Cells were washed with Opti-MEM, and transfection mixture was added. The cells were incubated for 6 h after transfection, washed, and cultured for 24 h in L-Ca medium.

### 2.6. Western Blot Analysis

After treatment with experimental compounds, HaCaT cells were harvested for Western blot analysis. Cells were briefly lysed with RIPA Buffer (Wako Co., Osaka, Japan) supplemented with cOmplete^TM^ (Roche, Basel, Switzerland) and PhosSTOP phosphatase inhibitor cocktail (Roche), and supernatant protein concentrations were determined using a bicinchoninate protein assay kit (Nacalai Tesque, Inc., Kyoto, Japan). Equal amounts of proteins (10 μg) were separated by sodium dodecyl sulfate polyacrylamide gel electrophoresis (SDSPAGE, WAKO SuperSepTMAce, 5–20%, Wako Co.), transferred to polyvinylidene fluoride (PVDF) membranes (Millipore Corp, Burlington, MA, USA) using a Trans-Blot SD Semi-Dry Electrophoretic Transfer Cell (Bio-Rad, Beerkeley, CA, USA), blocked with Blocking One (Nacalai Tesque, Inc.) for 1 h, and incubated overnight at 4 °C with primary antibodies: anti-p-c-Fos (1:2000), anti-c-Fos (1:2000), anti-p-p38MAPK (1:2000), anti-p38MAPK (1:2000), anti-involucrin (1:2000) anti-c-Fos (1:2000), or anti-β-actin (1:3000) diluted in 10% Blocking One Solution with 0.05% Tween 20 and 50 mM Tris-buffered saline (TBST). The following day, each membrane was washed with TBST three times, for 10 min each, and incubated with the corresponding horseradish peroxidase-conjugated secondary antibodies (1:2000 to 1:4000) for 1 h at room temperature. The membrane was then treated with chemiluminescent reagent (Chemi-Lumi One Super, Nacalai Tesque, Inc.). Antibody-bound protein bands were detected using a Syngene G: BOX Chemi XT4 fluorescence and chemiluminescence gel imaging system (Syngene). The relative intensities of developed bands were quantified using a commercial software (JustTLC) and normalized to the loading control.

### 2.7. Rac1 and RhoA Activation Assay

Cells were treated with kCer, histamine, or Rac-1 inhibitor (NSC237669). Cell lysates were treated with Rhotekin-RBD or PAK-RBD affinity beads from a small G-protein activation assay kit (Cytoskeleton, Inc., K035-S and BK036-S). After bound GTPase proteins were eluted from pelleted beads in SDS buffer, samples were subjected to SDS-PAGE and probed with RhoA or Rac-1 specific antibodies using a Western blot.

### 2.8. [3.H] Dihydrosphingosine (DHS) Labeling Assay

Labeling cellular sphingolipids in HaCaT cells using [5-^3^H] DHS (American Radiolabeled Chemicals, St. Louis, MO, USA), lipid extractions, and lipid separation by TLC were performed as described previously [18]. Lipid separation by normal-phase TLC was performed using Silica Gel 60 HPTLC plates (Merck) with 1-butanol/acetic acid/water (3:1:1, *v*/*v*) as the solvent system. Labeled lipids were detected using a bioimaging analyzer (BAS-2500, GE Healthcare Life Sciences, Tokyo, Japan).

### 2.9. LC-MS/MS Analysis

Lipids were extracted from cell pellets scraped from culture dishes using the Bligh and Dyer method with 0.1 nmol internal standard (C17 ceramide, C12 GlcCer, C17 SM) and glycerolipids were then degraded with NaOH. Sphingolipids were extracted with chloroform and dried under a nitrogen stream. The sphingolipid-enriched pellets were dissolved in acetonitrile: methanol (19:1 *v*/*v*) and subjected to LC-MS/MS analysis. LC-MS/MS analysis was conducted using a Triple TOF 5600 system (AB SCIEX, Foster City, CA, USA) equipped with an InertSustain NH2 column (diameter 2.1 mm, length 100 mm, particle size 5 μm, GL Science, Tokyo, Japan) in electrospray ionization (ESI)-positive mode as described previously [19].

### 2.10. Quantitative RT-PCR (qPCR)

Total RNA was extracted using PureLink RNA Mini Kit (ThermoFisher Scientific, Waltham, MA, USA). Total RNA concentration was measured using a NanoDrop micro-spectrophotometer, and the aliquots were stored at −80 °C until use. Total RNA was reverse-transcribed to cDNA using PrimerScript RT Master Mix (Takara Bio, Shiga, Japan). 

Real-time PCR was conducted using a TB Green Premix ExTaq II kit (Takara Bio, San Jose, CA, USA) and a Mx3000 QPCR System (Agilent Technologies, Santa Clara, CA, USA) according to the manufacturer’s instructions. The primer pair for involucrin was: forward (5′-GGTCCAAGACATTCAACCAGCC-3′) and reverse (5′-TCTGGACACTGCGGGTGGTTAT-3′), CerS3 forward (5′-CATGATCTTGCAGGTCCTTCACC-3′) and reverse (5′-CTCGTCATCACTCCTCACATCC-3′), and GAPDH forward (5′- GTCTCCTCTGACTTCAACAGCG-3′) and reverse (5′- ACCACCCTGTTGCTGTAGCCAA-3′). The PCR reaction conditions were as follows: 1 cycle at 95 °C for 10 s, then 40 cycles at 95 °C for 5 s, at 60 °C for 30 s, at 95 °C for 1 min, at 60 °C for 30 s, and at 95 °C for 30 s. Relative mRNA levels of involucrin and CerS3 were normalized by GAPDH.

### 2.11. Statistical Analysis

The number (n) in each experimental condition is specified in the Figure legends. Data were analyzed statistically using Prism 5.0 (GraphPad, San Diego, CA, USA). For comparison of two experimental conditions, statistical analysis was performed using a paired Student’s t-test. For comparison of three or more conditions, statistical analysis was conducted using a one-way analysis of variance (ANOVA) followed by Tukey’s multiple comparison post-test and Dunnett’s test. *p* < 0.01 was considered statistically significant. *p*-value ranges are indicated in the Figure legends. 

## 3. Results

### 3.1. kCer-Induced HaCaT Cell Migration and Differentiation

Migration of HaCaT cells was much more robust under H-Ca differentiation conditions than under L-Ca undifferentiation conditions (Figure 2(A1,A2,B1,B2)). kCer treatment not only inhibited cell migration in most HaCaT cells, but also decreased cell differentiation of the few migrated cells (Figure 2(B9,B10)). Sema3A treatment had a similar effect to that of kCer (Figure 2(B5,B6)). On the other hand, histamine treatment enhanced cell migration in most cells in both H-Ca and L-Ca conditions (Figure 2(A3,A4,B3,B4)). kCer co-treatment reversed histamine-induced cell migration and undifferentiation (Figure 2(B11,B12)). Sema3A treatment also reversed histamine-induced cellular behavior (Figure 2(B7,B8)). Taken together, these findings suggested that kCer-induced cell differentiation was similar to that of Sema3A in the Sema3A-Nrp1 receptor pathway (Figure 2C).

In the presence of histamine, kCer induced dose-dependent differentiation under H-Ca conditions (Appendix A). In addition, the differentiation activity of kCer was specific to this species of ceramide, as other ceramide species (kGlcCer or animal type ceramides such as C16Cer, C18Cer, and C24Cer) did not induce differentiation (Appendix A).

### 3.2. kCer Activation of Cell Differentiation Pathways

To determine if kCer-induced differentiation was concomitant with activation of the p38MAPK pathway, we performed Western blotting to measure involucrin, p38, p-p38, c-Fos, and p-c-Fos levels. kCer dose-dependently (0, 1, and 10 μM) increased p38 and c-Fos phosphorylation, followed by increased involucrin levels (Figure 3A and Appendix A). The dose dependency of kCer-induced p38MAPK activation in the presence of histamine occurred under both culture conditions, including both L-Ca and H-Ca (Appendix A). 

To identify the responsible p38MAPK isoform, cells were co-treated with kCer and a p38MAPK inhibitor (SB203580, BIRB796, or VX-745). These inhibitors did not affect the expression levels of p38, c-Fos, or β-actin, and p-p38 and p-c-Fos levels were inhibited only by BIRB796, an inhibitor of p38α/β/γ/δ (Figure 3B and Appendix A). In addition to the inhibitor experiment, siRNA interference further underscored the essential role of p38δ H-Ca-induced cell differentiation (Appendix A).

kCer-induced differentiation was inhibited by siRNA targeting p38δ and PLCβ, but not PKCα in H-Ca cells (Appendix A). siRNA interference had similar effects in L-Ca cells treated with kCer (Appendix A). Histamine suppression of differentiation was not changed by siRNA targeting p38δ, PLCβ, or PKCα (Appendix A). By contrast, histamine treatment induced differentiation under H-Ca conditions, but gene silencing with siRNA targeting p38δ, PLCβ, and PKCα suppressed the combined effect of H-Ca medium and histamine (Appendix A). Neither kCer nor histamine treatment changed the suppression of cell differentiation by gene silencing of p38δ, PKCα, and PLCβ in L-Ca conditions (Appendix A). On the other hand, H-Ca medium induced similar effects to that of kCer without histamine (Appendix A). This effect was confirmed by the finding that PKCα was not suppressed by siRNA targeting PKCα (Appendix A).

### 3.3. Effect of kCer on the Balance of Rac1/RhoA Activities

kCer-induced Rac-1 activation is followed by cofilin phosphorylation and inhibition of cell migration [12] under short-term L-Ca conditions. A coordinated balance of Rac1 and RhoA regulates cell shape, cell motility [20], and keratinocyte differentiation [21]. 

To examine the specific contributions of kCer and histamine on coordination of Rac1/RhoA activities, as illustrated in Figure 4A, HaCaT cells were treated with either kCer or histamine in L-Ca medium, and subjected to a small G-protein activation assay. Histamine decreased Rac1 activity and increased RhoA activity (Appendix A). Contrastingly, Sema3A and kCer increased Rac1 activity and decreased RhoA activity. Sema3A and kCer also reversed the effects of histamine in cells co-treated with histamine and either Sema3A or kCer. Rac1 and RhoA were activated by the GDP and GTP exchange proteins, respectively. To determine if the effect of kCer was due to Rac1 activation or RhoA inhibition, cells were co-treated with kCer and the Rac1 inhibitor NSC23766 and subjected to the Rac1/RhoA activation assay as illustrated in Figure 4B. NSC23766 inhibited Rac1 activity, which was reversed by kCer co-treatment (graph below Figure 4B).

### 3.4. Effect of kCer on Histamine-Induced RhoA Activation

We speculated that the RhoA signaling pathway could regulate p38MAPK activation in long-term L-Ca conditions. Previously, we demonstrated that neither kCer nor Sema3A are involved in histamine-GPCR activation [12]. We next examined how histamine-induced RhoA activation contributed to cell undifferentiation. We hypothesized that downstream p38MAPK could be suppressed Rock/RhoA kinase. To test this hypothesis, cells under long-term L-Ca conditions were treated with vehicle, histamine, or kCer in monotreatment or in co-treatment with the ROCK/RhoA kinase inhibitor Y27632. Neither Y27632 nor histamine-activated p38MAPK, but both Y27632 and histamine increased p38MAPK activity in cells co-treated with 0.1 μM Y27632 (not significant) and 1.0 μM Y27632 (* *p* < 0.01, Figure 5A,B). Co-treatment with Y27632 and kCer had an additive effect on p38MAPK activity (* *p* < 0.01). These findings demonstrated that inhibition of the RhoA pathway is followed by p38MAPK pathway activation In summary, kCer activates the Sema3A pathway, followed by Rac1 activation and subsequently p38MAPK activation. Due to the feedback mechanism between Rac1 and RhoA, RhoA is suppressed, followed by histamine-induced ROCK/RhoA kinase suppression, leading to p38MAPK activation. Here we shows our proposed model for downstream pathway crosstalk (Figure 5C).

### 3.5. Effect of Cell Differentiation on Sphingolipid Metabolism 

Sphingolipid metabolism was examined by metabolic labeling under short-term L-Ca conditions and by LC-MS analysis under long-term L-Ca conditions. Treatment with kCer in L-Ca medium encouraged biosynthesis of GlcCer, long-chain Cer, very long-chain Cer, and other GSL SM species, compared with kCer in H-Ca medium (Appendix A). qPCR analysis demonstrated that CerS3 was not increased, but that involucrin was non-significantly increased under short-term L-Ca conditions (NS, Appendix A). On the other hand, sphingolipid metabolism was affected under long-term L-Ca conditions (Appendix A and Appendix A). Total ceramide was increased by kCer treatment but decreased by histamine treatment. Total SM had a similar profile to that of total ceramides. 

## 4. Discussion

Keratinocyte migration is an important process in the terminal differentiation of keratinocytes. Our prior study demonstrated that kCer regulates keratinocyte migration via a Sema3A-like repulsion mechanism [12]. 

Extracellular calcium levels play an important modulatory role in activation of epidermal keratinocytes [22]. Primary keratinocytes cultured in vitro at low calcium concentrations retain a basal phenotype, while calcium concentrations > 0.1 mM induce keratinocyte differentiation. Naturally immortalized human HaCaT cells can be grown in conventional medium and retain keratinocyte differentiation features, which can be reversed by the long-term culture in low calcium conditions [15]. 

The present study indicates that kCer promotes HaCaT cell differentiation after short-term and long-term culture in L-Ca medium. The keratinocyte growth and differentiation switch, which is tightly regulated by several mechanisms, is generally associated with decreased proliferation, cell cycle arrest, and increased levels of epidermal differentiation markers such as involucrin and p-p38MAPK [23]. c-Fos protein is specifically activated by phosphorylation during differentiation of human keratinocytes [24]. In the present study, kCer-induced HaCaT cell differentiation was accompanied by well-known differentiation features of keratinocytes such as elevation of involucrin protein and phosphorylation of p38MAPK and c-Fos. The epidermal skin layer provides a protective barrier from the outside environment. Cell differentiation occurs in migrating cells after stimulation with kCer (Figure 2). kCer treatment had a similar effect on HaCaT cells to that of H-Ca medium, inducing expression of involucrin and activation of p38MAPK (Figure 3) [25].

In the present study, in addition to determining if kCer treatment induced involucrin expression, we examined whether kCer affected skin permeability-related lipid components. kCer non-significantly increased involucrum mRNA (*p* > 0.01), and did not affect CerS3 mRNA levels (Appendix A). We also demonstrated that kCer stimulated biosynthesis of GlcCer and Cer with long/very long-chain fatty acids in L-Ca medium (Appendix A). kCer treatment increased total Cer and CMH, while histamine treatment decreased total Cer and CMH (Appendix A). 

Nrp1 is upregulated in primary keratinocytes in response to differentiating media [13]. Sema3A represses neurogenesis via Nrp1, and also plays an important role in keratinocyte migration and differentiation [26,27,28]. Our previous study demonstrated that Sema3A and kCer do not affect histamine-GPCR-mediated cell migration and undifferentiation [12]. However, considering that kCer and Sema3A still suppressed histamine-induced cell migration and undifferentiation, we postulated that the Sema3A/Nrp1 pathway could regulate histamine-induced effects via crosstalk between intracellular pathways downstream of Histamine-GPCR and Sema3A-Nrp1 interactions. In the present study, we hypothesized that kCer and histamine could modulate the balance between Rac1 and RhoA in a shared regulatory mechanism of cell migration and cell differentiation. The Rho family of small GTPases is involved in the regulation of neuronal morphology, cell shape and invagination, epithelial invagination, and phagosome formation during phagocytosis. As demonstrated by Bolado-Carrancio et al. [20], Rac and Rho exert mutual suppressive influences in breast cancer cells, so we postulated that the balance of Rac1 and RhoA activities could regulate migration and differentiation in HaCaT cells. It has also been reported [29] that cell migration of HaCaT cells is induced by Rac1 activation and RhoA inhibition by a certain plant pigment. Migrating cells must coordinate distinct leading and trailing dynamics, but the underlying mechanism is unclear. The coordinated balance of Rac1 and RhoA as minimal autonomous biochemical machinery for this dynamic coordination and cell movement needs to be further elucidated. In the leading edge of moving cells, RhoA activates Rac1 by inhibiting formin-1 (DIA) and inhibits Rac1 activity by activating ROCK. On the other hand, in the cell body and trailing edge of moving cells, activated Rac1 inhibits RhoA by activating PAK [20]. Our data suggest that the latter mechanism is responsible for the suppressive effects of kCer and Sema3A on histamine-induced cell migration and undifferentiation of keratinocytes. We demonstrated that kCer-induced Rac1 activation/RhoA inhibition was the responsible downstream intracellular mechanism of HaCaT cell differentiation. We also demonstrated that kCer-induced suppression of keratinocyte migration via the Sema3A pathway inhibits the histamine-activated GPCR pathway via crosstalk of downstream signaling cascades (Figure 6). The second effect of kCer could stimulate keratinocyte differentiation. Histamine is reported to enhance c-Fos expression, NGF production, and membrane PKC activation in human foreskin keratinocytes [30]. These effects could be mediated by H1-receptor-induced signaling, namely, the PLC-PKC-ERK cascade. By contrast, histamine did not enhance c-Fos expression or PKC activation in HaCaT cells (Appendix A). This discrepancy may be due to cell type-specific effects, in which histamine regulates the PLC-PKC-ERK cascade in foreskin keratinocytes and the RhoA-ROCK/RhoA kinase-p38MAPK cascade in HaCaT cells. 

Histamine activates RhoA (Figure 4), and active RhoA is inhibited by the ROCK/RhoA kinase inhibitor Y27632 (Figure 5). Although the mechanism is still unknown (Figure 5C), histamine ROCK/RhoA kinase activation could suppress p38MAPK phosphorylation. Contrastingly, kCer-induced Rac1 activation suppresses RhoA activity. We speculate that suppression of ROCK/RhoA kinase activity could stimulate downstream phosphorylation of p38MAPK (Figure 5C). A recent report demonstrated that Y27632 inhibited myofibroblast differentiation via RhoA mediation of the ERK and p38 MAPK signaling pathways [31]. On the other hand, Rac1 and Cdc42 couple to and regulate p38MAPK and are regulated by Pak1 [32,33]. RhoA and Rac1 are likely associated with p38MAPK in pathway crosstalk.

HaCaT keratinocytes possessed a characteristic PKC isoform pattern (PKCα, β, γ, δ, ε, η, θ, ζ), which is altered during proliferation and differentiation [34]. PKCα siRNA inhibited H-Ca-induced p38MAPK activation (Appendix A). kCer-induced differentiation was accompanied by PKCα, which could regulate p38MAPK activity. Four p38MAPK isoforms (p38α, β, γ, and δ) regulate a diverse array of cellular processes. Among the four described p38 isoforms, p38α, β, and δ are expressed in keratinocytes [35]. However, the effects of individual p38 isoforms on keratinocyte function remain incompletely understood. In the present study, we characterized the role of p38 MAPKδ as a regulator of keratinocyte differentiation by use of isoform-specific inhibitors: VX745, (p38α, p38β, p38γ), BIRB796 (p38α, p38β, p38γ, p38δ), and SB203580 (p38α, p38β).

The MAPK signaling pathway has critical roles in epidermal differentiation and skin barrier function [25,36,37,38]. p38δ is stimulated by okadaic acid, and is a potent inhibitor protein phosphatase 2A, which induces keratinocyte differentiation [39]. The MAPK signaling pathways can integrate and mediate multiple signals involved in keratinocyte differentiation, including calcium signaling. Inhibition of the p38 MAPK pathway in keratinocytes could interfere with keratinocyte differentiation. p-p38MAPK levels increase during the differentiation process of keratinocytes. In addition, p38MAPKδ activation is followed by phosphorylation of c-Fos (Figure 2), a downstream transcription factor activator, and expression of involucrin, a major component of the epidermis during late keratinocyte differentiation stages (Figure 6). 

kCer is Sema3A-like ligand that binds and activates the Sema3A receptor Nrp1, as we previously reported [5]. Sema3A binds the a1-a2 domain of Nrp1. kCer also binds a site A near the Sema3A binding region of a1, followed by signal transferring to the b1-b2 domain after active formation by IDR-like flexibility regulation due to the distance between the a1 and a2 molecules. Nrp1 also functions as a cell surface receptor for multiple ligands that bind to its b1-b2 domain through a C-terminal R/KXXR/K motif, similar to RPARPAR which is often found in the carboxyl (C)-terminal regions of many growth factors and other signaling molecules. Such a basic motif of their C-terminal regions of these molecules has been identified as C-end Rule or CendR motif [40]. Recently, the CendR motif was also found at the furin cleavage site of the SARS-CoV-2 spike protein and to be associated with the virus infection [41]. Peptides with this motif (CendR peptides) are taken up into cells by endocytosis [42]. These ligands include the vacular endothelial growth factor (VEGF)-A isoform VEGF-A_165_, transforming growth factor β, and Sema3A. Sema3A binding to Nrp1 recruits plexin A1, another receptor molecule joined to the receptor complex. These complex molecules have additional receptors to trigger downstream signaling events. Thus, the RPARPAR peptide becomes a triggering molecule for endocytosed intracellular Nrp1 receptors. CendR endocytosis and subsequent intercellular transport of CendR cargo are stimulated by nutrient depletion. 

In the present study, kCer-induced HaCaT cell differentiation by PKCα and PLCβ activation occurred via Ca^2+^ mobilization from intracellular Ca^2+^ stores. CendR is a bulk transport pathway, so our results suggest that the role of kCer transport into intracellular Ca^2+^ stores could be exploited via strategies such as CendR-enhanced drug delivery. The proposed mechanism is summarized in Figure 6. 

## 5. Conclusions

kCer differentiates keratinocytes by activating the Sema3A-Nrp1 signaling pathway. kCer competes with the downstream signaling events of histamine-activated GPCR and Sema3A-Nrp1. These findings highlight the cascade crosstalk effect by kCer. In future study, we need to clarify the mechanism to make the reasons for the functional difference of kCer-induced activations in keratinocyte and neuron.

## Figures and Tables

**Figure 1 biology-11-00121-f001:**
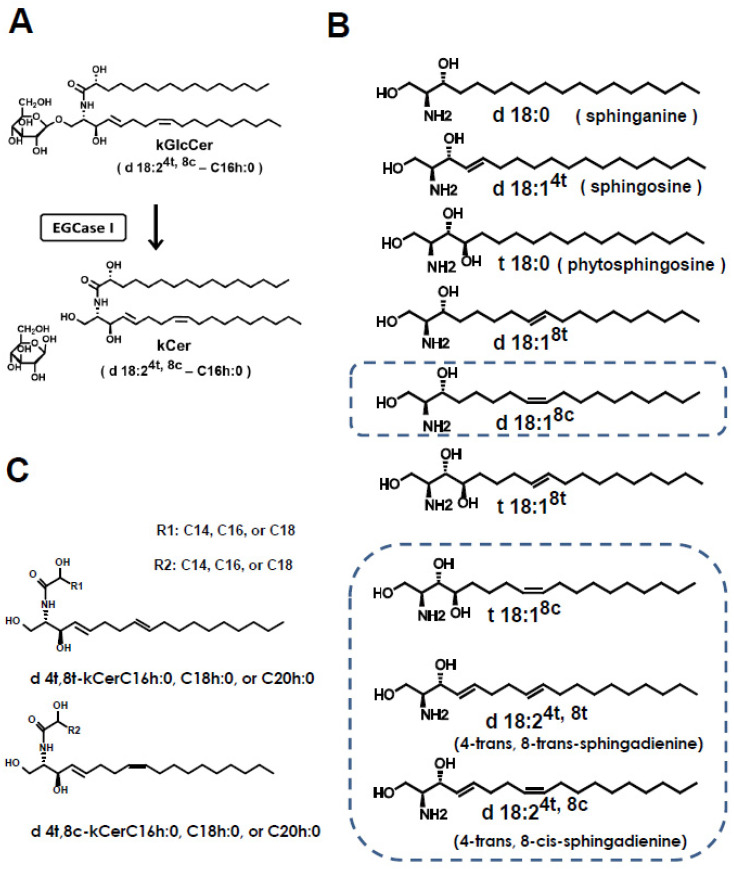
Chemical structure of ceramide species and EGCase Ι-catalyzed degucosylation for kCer preparation. (**A**) The EGCase reaction of kGlcCer. Plant-type ceramides can be prepared by treating plant-type GlcCer with EGCase I. A major molecular species of kCer (d18:2^4t,8c^-C16h:0) is shown. (**B**) Primary long-chain bases found in plants. The long-chain bases included in konjac ceramide (kCer) produced by EGCase treatment are indicated by a dotted rectangle. (**C**) kCer molecular species generated by EGCase treatment of kGlcCer. The carbon chain length of each hydroxyl fatty acid (C14, C16 and C20) is shown in R1 and R2, respectively. Figure 1 is referred from [5], and used the copyright permission of the corresponding author. The nomenclature for long-chain bases and ceramides follows the recommendations of the IUPAC-IUBMB Joint Commission.

**Figure 2 biology-11-00121-f002:**
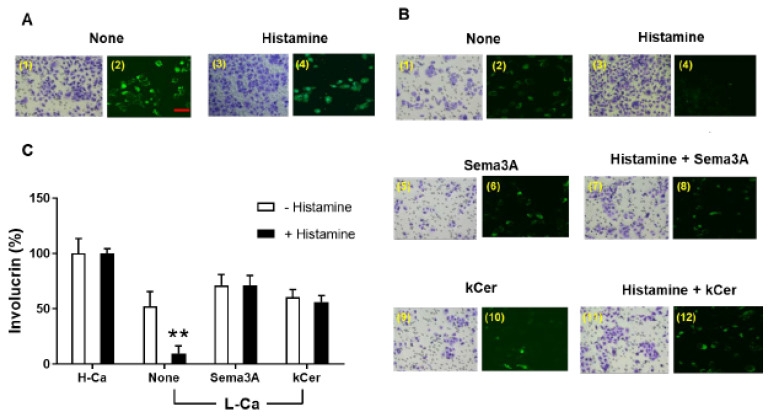
kCer-induced HaCaT cell differentiation and migration. Short-term cell culture was used for the HaCaT cell migration assay. Cell migration activity was determined quantitatively using ImageJ after GIEMSA’S AZUR EOSIN Methylene Blue-staining of the transwell filter membrane (left images in (**A**,**B**)). Cells that migrated to the opposite side of the transwell membrane were immunostained with anti-involucrin antibody (right image in (**A**,**B**)). (**C**) The differentiation effect of Sema3A (100 nM) and kCer (25 μM) on HaCaT cells after short-term culture was compared in the presence or absence of 10 μM histamine and quantified using Image J. Data are presented as means ± SD (*n* = 3, ** *p* < 0.01). Scale bar = 100 μm.

**Figure 3 biology-11-00121-f003:**
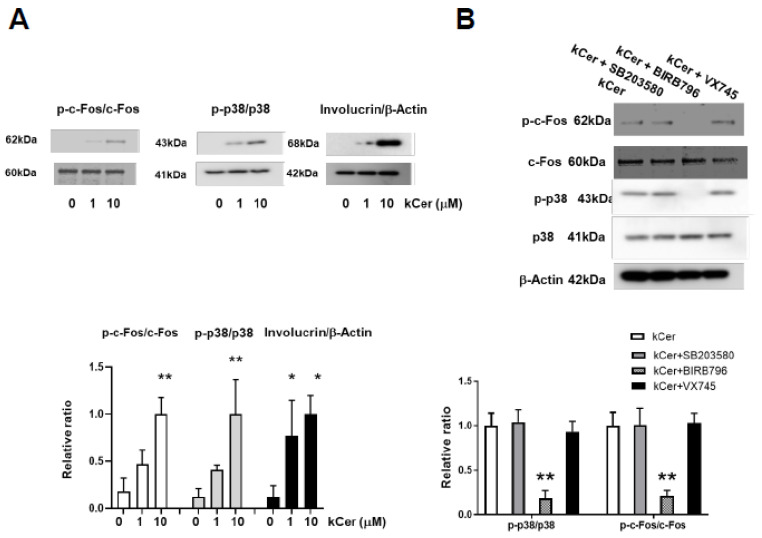
Western blot analysis of kCer-induced HaCaT cell differentiation and signaling pathway activation. (**A**) Effects of treatments (0, 1, and 10 μM kCer) on cell protein (c-Fos, p-c-Fos, p38, and p-p38) expression were measured by Western blot analysis. The blotted bands were analyzed by Image J and graphed as shown below. Bars represent the relative ratios of p-c-Fos/c-Fos, p-p38/p38, and involucrin/β-Actin. (**B**) Effects of treatment with 10 μM kCer treatment ±1.0 μM of the p38MAPK inhibitor SB203580 (p38α/p38β), BIRB796 (p38α/p38β/p38γ/p38δ), or VX745 (p38α/p38β/p38δ), respectively. Cell lysates were subjected to immunoblot analysis, followed by detection of p-c-Fos, c-Fos, p-p38, p38, and β-Actin. Immunoblot bands were analyzed by Image J, and the relative ratios of p-p38/p38 and p-c-Fos/c-Fos are graphed below. Data are presented as means ± SD (*n* = 3, ** *p* < 0.01, * *p* < 0.05).

**Figure 4 biology-11-00121-f004:**
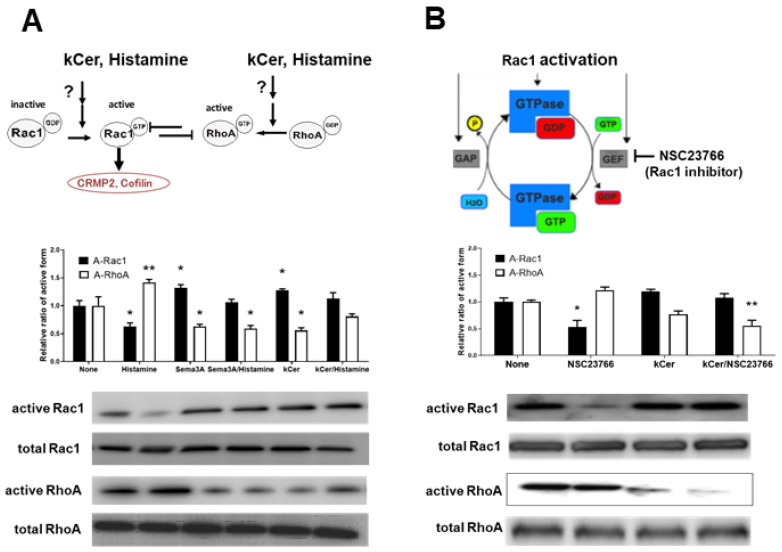
Effects of kCer and histamine on mutual inhibition of the Rac1 and RhoA activation mechanism. (**A**) Two possible working sites of kCer and histamine potentially affect mutual inhibition of Rac1 and RhoA activation, as illustrated in the above chart. In the presence of kCer or histamine, Rac1 and RhoA activation was examined using a G-protein activation assay kit. Western blot band densities were quantified by Image J, and the relative ratio of active/total Rac1(A-Rac1) and active/total RhoA (A-RhoA) are graphed below. Data are presented as the means ± SD (*n* = 3). * *p* < 0.05, ** *p* < 0.01 vs. no additive (None), one-way ANOVA followed by Tukey’s multiple comparison post-test and Dunnett’s test. (**B**) Illustration of Rac1 activation and the Rac1-specific inhibitor (NSC237669) that prevents GTPase-activation via guanine nucleotide exchange factor (GEF). Active Rac1 is also inactivated by GTPase-activation protein (GAP). Effects of NSC237669 on Rac1 and RhoA activation were examined in the presence or absence of kCer using a G-protein activation assay kit as described above. Data are presented as the means ± SD (*n* = 3), followed by statistical analysis as described in (**A**).

**Figure 5 biology-11-00121-f005:**
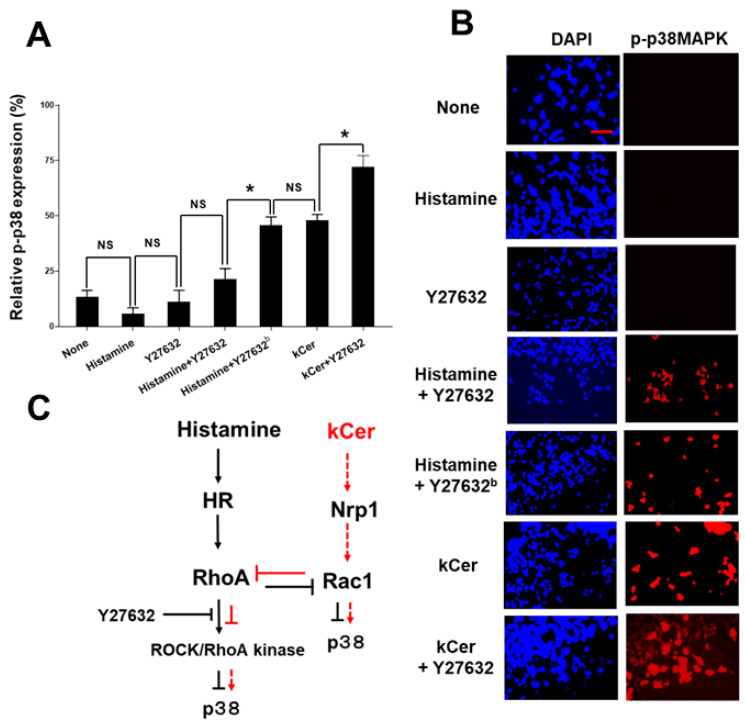
Downstream pathway crosstalk between histamine-related (HR) signaling and kCer-related Nrp1 signaling. (**A**) Effects of Y27632 on p38MAPK activation induced by treatment with histamine and kCer. HaCaT cells were undifferentiated by long-term cell culture in L-Ca medium. Cells were then treated with histamine (10 μM), Y27632 (0.1 μM), Y27632^b^ (1.0 μM), or kCer (10 μM). The cells were subjected to a p-p38MAPK cell-based translocation assay kit. The results are presented as the means and standard deviations of three individual experiments. Statistical significance (* *p* < 0.01 vs. the represented treatment), not significant (NS), respectively, one-way ANOVA followed by Tukey’s multiple comparison post-test. (**B**) Representative images of pp38MAPK activation in experiments using Y27632, kCer, and histamine. Scale bar = 100 μm. (**C**) Proposed model for downstream pathway crosstalk of histamine and kCer.

**Figure 6 biology-11-00121-f006:**
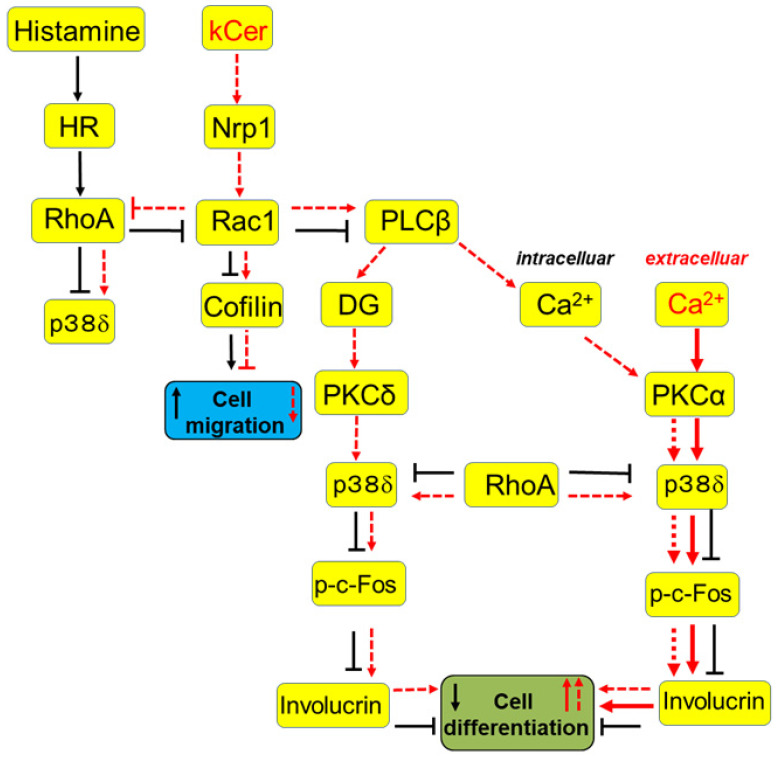
Proposed mechanism of signaling crosstalk between Sema3A-Nrp1 and HR pathways. Results of stimuli by histamine and high extracellular Ca^2+^ are reflected in the possible mechanisms for kCer regulation of cell migration and differentiation. HR, histamine receptor; Nrp1, neuropilin1; DG, diacylglycerol. Black arrows, activations via histamine; broken-lined red arrows, activations by kCer; thick-lined red arrows, activations by high Ca^2+^; 
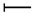
, suppression via histamine.

## Data Availability

The data presented in this study are available in the article.

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
