# Peer review of "Konjac Ceramide (kCer)-Mediated Signal Transduction of the Sema3A Pathway Promotes HaCaT Keratinocyte Differentiation"

_biology, 2022, doi:10.3390/biology11010121_

Round 1
Reviewer 1 Report
Manuscript by Usuki et al., Konjac Ceramide (kCer)-mediated signal transduction of the Sema3A pathway promotes HaCaT keratinocyte differentiation demonstrates the role of kCer on cell migration and cell differentiation, and this mediated by a cascade of crosstalk between pathways downstream Nrp1 and histamine-GPCR.
Comments:
Overall manuscript is well written, and data presented with respective controls. The conclusions are nicely drawn, and results are well explained.
There are few minor issues that needs attention:
- The quality of Fig 5 B is not good. Authors should also include the phase-contrast micrograph of the cells.
- There seems to be a typo in Fig 6 model, it should be “intracellular” & “extracellular”
- Typo line# 361 and 362., where is the Fig 8C (as mentioned) in the manuscript.
Author Response
Authors’ Response to the Reviewers’ Comments
Journal: biology (ISSN 2079-7737)
Manuscript #: biology-1527303
Title of Paper: Konjac Ceramide (kCer)-mediated signal transduction of the Sema3A
pathway promotes HaCaT keratinocyte differentiation
Authors: Seigo Usuki, Noriko Tamura, Tomohiro Tamura, Kohei Yuyama,
Daisuke Mikami, Katsuyuki Mukai, and Yasuyuki Igarashi
We appreciate very much the interest of the editor and reviewers in reviewing this manuscript. We have addressed each of their comments and the changes are shown by the part corrected in red in the manuscript and the supplement. The revised manuscript has been checked by English Proofreading Service according to both reviewers’ suggestion.
We believe that the revised version can meet the journal’s publication requirements.
Responses to comments from Reviewer 1
- The quality of Fig 5 B is not good. Authors should also include the phase-contrast
micrograph of the cells.
Response: We are sorry to tell you that we do not store clear phase micrograph photos. Instead of it, we replaced it by a higher quality of the fluorescence photos.
- There seems to be a typo in Fig 6 model, it should be “intracellular” & “extracellular”
Response: We appreciate the reviewer. It was replaced by the words, “intracellular” & “extracellular”
- Typo line# 361 and 362., where is the Fig 8C (as mentioned) in the manuscript.
Response: In line #361 and 362, “Figure 8A” and “Figure 8C” have been changed to “Figure S8A” and “Figure S8C”.
Reviewer 2 Report
In this study, Usuki et al. investigated the role of Konjac ceramide (kCer) regulated signaling mechanism of Sema3A pathway to identify its role in HaCaT keratinocyte differentiation. In my opinion, the study is planned nicely, and findings are quite interesting. The data looks robust, and manuscript is written well. However, I have several comments that can be considered for improving the quality of manuscript.
- How was the expression level in qRT-PCR results calculated? Which sample was used as reference sample for calculating relative quantification if 2-ΔΔCT method was used?
- There are several typographical, and errors in protein or gene names in the figures and text. The authors need to check it carefully.
- It would be good to include some future prospects of the study particularly in the conclusion section.
Author Response
We appreciate very much the interest of the editor and reviewers in reviewing this manuscript. We have addressed each of their comments and the changes are shown by the part corrected in red in the manuscript and the supplement. The revised manuscript has been checked by English Proofreading Service according to both reviewers’ suggestion.
We believe that the revised version can meet the journal’s publication requirements.
Responses to comments from Reviewer 2
- How was the expression level in qRT-PCR results calculated? Which sample was used
as reference sample for calculating relative quantification if 2-ΔΔCT method was used?
Response: We used GAPDH as a reference to quantify relative mRNA levels of involucrin and CerS3. We have added a following sentence (page 7, line 205), "Relative mRNA levels of involucrin and CerS3 were normalized by GAPDH”.
We also added a sentence (supplement page 10, line 137), “Data were expressed as relative ratio of L-Ca samples values”.
- There are several typographical, and errors in protein or gene names in the figures and
text. The authors need to check it carefully.
Response: According to the reviewer’s comment, gene names have been checked. Ceramide synthase 3 has been written as “CerS3” (page 7, line 200; page 12, line 331).
- It would be good to include some future prospects of the study particularly in the
conclusion section.
Response: We agreed the reviewer’s comment. We have mentioned it in the conclusion section (page 16, line 466 to 468), “These findings highlight the cascade crosstalk effect by kCer. In future study, we need to clarify the mechanism to make the reasons for the functional difference of kCer-induced activations in keratinocyte and neuron”.